# Liver Resection Using Saline-Linked Radiofrequency Technology in an Infant with Congenital Hepatoblastoma

**DOI:** 10.3390/children9030418

**Published:** 2022-03-15

**Authors:** Giovanni Torino, Michele Ilari, Edoardo Bindi, Francesca Mariscoli, Giovanni Cobellis

**Affiliations:** 1Pediatric Surgery Unit, Salesi Children’s Hospital, 60123 Ancona, Italy; giovannitorino1@libero.it (G.T.); michele.ilari@ospedaliriuniti.marche.it (M.I.); francesca.mariscoli@ospedaliriuniti.marche.it (F.M.); giovanni.cobellis@ospedaliriuniti.marche.it (G.C.); 2Faculty of Medicine and Surgery, University Politecnica of Marche, Via Tronto, 10/a, 60126 Ancona, Italy

**Keywords:** transcollation, floating ball, dissecting sealer, bloodless resection, pediatric age

## Abstract

We herein report a case of giant congenital hepatoblastoma in a 3-month-old male treated with neoadjuvant chemotherapy and hepatic resection. After considerable reduction of the tumor with chemotherapy, a right bloodless hemihepatectomy using saline-linked radiofrequency technology (SLRT) and without clamping of the hepatic pedicle was performed. Intraoperative blood loss was minimal, and consequently, no blood transfusions were required. The surgery lasted 140 min, and SLRT was used for a total of 60 min. No complications were observed during or after the surgery. In conclusion, congenital hepatoblastoma is a very rare cancer for which surgery is an essential therapeutic step, and in our presented case, we showed that SLRT allowed for a safe and effective bloodless liver resection.

## 1. Introduction

Malignant liver tumors represent approximately two-thirds of all liver masses and account for 1% of childhood tumors overall [1]. The most frequent tumors are hepatoblastoma (HBL) and hepatocellular carcinoma. In Western countries, approximately half of the liver tumors noted in childhood are HBLs [2].

Congenital HBL (cHBL) can be identified in utero, at birth, or during the first month of life. It is a very rare condition and has some important distinctive features compared with HBL diagnosed in children beyond neonatal age [3]. cHBL can present as an asymptomatic abdominal mass, and thus the diagnosis may be delayed because of the indolent and nonspecific presentation. Serum alpha-fetoprotein (AFP) has been found to be a useful marker in neonates presenting a hepatic mass (normal value of 0–6 ng/mL after 12 months of age). Nevertheless, in the first months of life, AFP levels can be significantly higher than this, even in the absence of a liver neoplasm, making it difficult to adequately use the AFP level as a specific marker for HBL. Thus, in the presence of a mass related to the liver and high levels of AFP, there is the need to proceed to an ultrasound-guided biopsy.

In the case of malignant liver tumors, the most effective treatment is surgical removal combined with chemotherapy (cisplatin, 5-fluorouracil, vincristine, carboplatin, and doxorubicin) before and/or after surgery. The goal of surgery is to completely remove the tumor by partial or complete resection of the liver with possible transplantation.

Over the past 20 years, technological advances have led to the development of specific devices for bloodless liver transection, such as ultrasonic cavitation devices, water jets, harmonic scalpels, and vessel-sealing devices. The Floating Ball, Dissecting Sealer, and Bipolar Sealer devices used for liver transection are based on saline-linked radiofrequency energy [4]. This energy allows for the transformation and fusion of collagenous fibers by remodeling their triple helix structure, a process that the manufacturer termed transcollation [5].

SLRT has already been described in adults, but to our knowledge, only three cases have been reported in the pediatric population, and three were older than 1 year old [6].

## 2. Case Report

Our case involves a male term infant with an abdominal mass detected by sonography at 36 weeks of gestation. The dimensions of the mass were approximately 6.8 × 7 × 9.1 cm, and it appeared to be associated with the liver. The differential diagnoses included a benign liver mass (hemangioma, hamartoma, or adenoma) or a malignant neoplasm (hepatoblastoma). The patient had an uneventful cesarean delivery in order to avoid rupture of the abdominal mass. The amniotic liquid was normal, and the birth weight was 3900 g. Upon abdominal examination, the infant had a palpable abdominal mass but was otherwise asymptomatic with no respiratory distress.

At 3 days of age, the patient was transferred to our unit. Blood tests were performed reporting increased levels of transaminase, lactic dehydrogenase, gamma-glutamyl transferase, and total bilirubin associated with increased levels of AFP (100.000 ng/mL vs. normal values of 0–50 ng/mL).

Ultrasound and magnetic resonance imaging (MRI) showed an abdominal mass of 6.5 × 7.1 × 9.5 cm related to the liver. Chest computerized axial tomography (CAT) and brain MRI were both normal.

At 7 days of age, the diagnosis of “epithelial fetal hepatoblastoma” was confirmed by open biopsy. The cancer was classified as third-stage per the SIOPEL PRETEXT protocol [2].

The case was discussed within our multidisciplinary team dealing with pediatric oncology. In agreement with our colleagues in oncology and in accordance with the SIOPEL protocol, we decided to start neoadjuvant chemotherapy with four subsequent cycles of cisplatin (70 mg/m^2^/day). At the end of the pharmacological treatment, a reduction in AFP and total bilirubin in the blood was observed. A new total-body computerized tomography (CT) showed a marked reduction in the mass (Figure 1) and confirmed the absence of systemic metastases.

At 3 months, the patient underwent a right hemihepatectomy by SLRT with transparenchymal access and without clamping of the hepatic pedicle (Pringle maneuver).

Intraoperative blood loss was minimal; no transfusions of blood products were required (bloodless resection). The total surgical time was 140 min, and SLRT was used intermittently for a total time of 60 min.

The postoperative course was free of noteworthy complications, and on the 2nd postoperative day, gradual stabilization of the liver enzyme levels was observed.

On the 7th postoperative day, the patient was discharged. The patient was completely asymptomatic and administered postoperative chemotherapy with two subsequent cycles of cisplatin. At 6 months follow-up, the total body MRI showed no recurrence of disease and hepatic compensatory hypertrophy of the left liver (Figure 2).

## 3. Discussion

cHBL is an extremely rare cancer. From 1966 to 2021, approximately 51 cases have been described in the literature [3,7,8,9,10,11,12,13,14]. This congenital form has some specific features that distinguish it from the form diagnosed beyond the neonatal period. A particular aspect of cHBL is its clinical presentation, which can be characterized by polyhydramnios and/or fetal hydrops associated with the death of the fetus, intra-abdominal hemorrhage due to the rupture of the tumor during labor or delivery, and severe respiratory distress after birth. This symptomatology is explained by the large dimensions that this tumor can reach during fetal life before being diagnosed. Notably, cHBL also tends to have a worse prognosis than HBL in older patients with differing metastatic patterns, namely bone and brain metastases in cHBL compared with lung metastasis in HBL in older patients [3]. This is explained by the lung bypass due to the physiological right–left shunt in the fetal circulation.

In the literature, recent data have highlighted the importance of preoperative chemotherapy in the treatment of hepatoblastoma [3,12]. In the case reported, the considerable dimensions of the cancer did not allow surgical removal, so preoperative chemotherapy was administered, which led to a significant reduction in the dimensions of the mass and allowed for subsequent surgical removal.

Liver resection is classified into major and minor hepatectomy according to the Couinaud classification [15]. Major hepatectomy is a resection of more than three segments. By contrast, the resection of three or fewer segments is considered minor hepatectomy. Massive operative hemorrhage is a risk associated with liver transection and particularly major hepatectomy. The number of hepatic segments resected and operative blood loss are the leading predictors of both perioperative morbidity and mortality. A reduction in both these factors significantly decreases perioperative mortality [16]. As a result, intraoperative prevention of blood loss is considered the gold standard of liver surgery. A possible strategy to reduce bleeding during transection of the liver is the Pringle maneuver (PM), but its continuous application for a total of more than 90 min may have deleterious effects on liver function [14]. In recent years, new instruments using different types of energy for the coagulation or sealing of vessels have been developed for liver bloodless transection. Among these new technologies, SLRT, as described in the literature for adults, permits good control of surgical blood loss without the PM, and was thus used in this case [4,17,18].

SLRT combines an electrosurgical generator for the delivery of radiofrequency energy and saline solution (0.9% NaCl) [4]. Devices utilizing SLRT facilitate hepatic resection through blunt dissection, hemostatic sealing, and coagulation of soft tissue (bloodless). To ensure a continuous flow of the saline solution, it is essential to avoid parenchymal charring, and the aspiration of excess saline solution is essential to avoid the dispersal of energy [5]. The capsule and parenchyma are incised along the section line through conventional devices, and then the operator continues the resection with SLRT. This technology seals vascular and biliary structures up to 3 mm in diameter by collagen fusion [5], and it permits precise and controlled tissue removal with no sticking, eschar, or smoke.

A bloodless right hemihepatectomy by SLRT was performed in the case reported. SLRT allowed excellent control of the intraoperative bleeding with insignificant blood loss without the PM. The good bleeding control during the operation allowed for a shorter anesthesia duration and minimized the overall surgical stress to the patient.

In conclusion, congenital hepatoblastoma is a very rare cancer for which surgery is an essential therapeutic step. In our case, SLRT allowed for a safe and effective bloodless major liver resection.


## Figures and Tables

**Figure 1 children-09-00418-f001:**
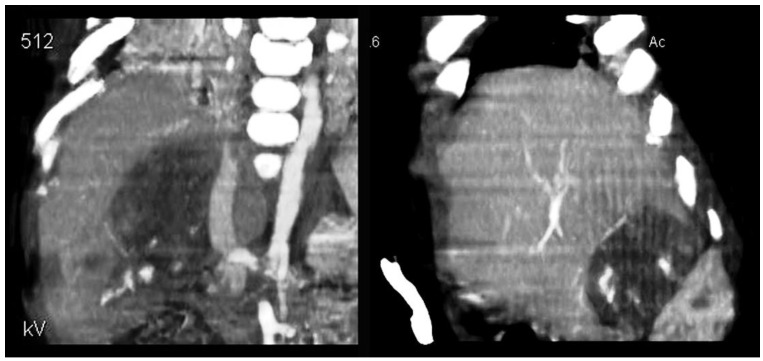
Legend—CT scan: marked reduction in the liver mass (image on the left before treatment) after chemotherapy (image on the right).

**Figure 2 children-09-00418-f002:**
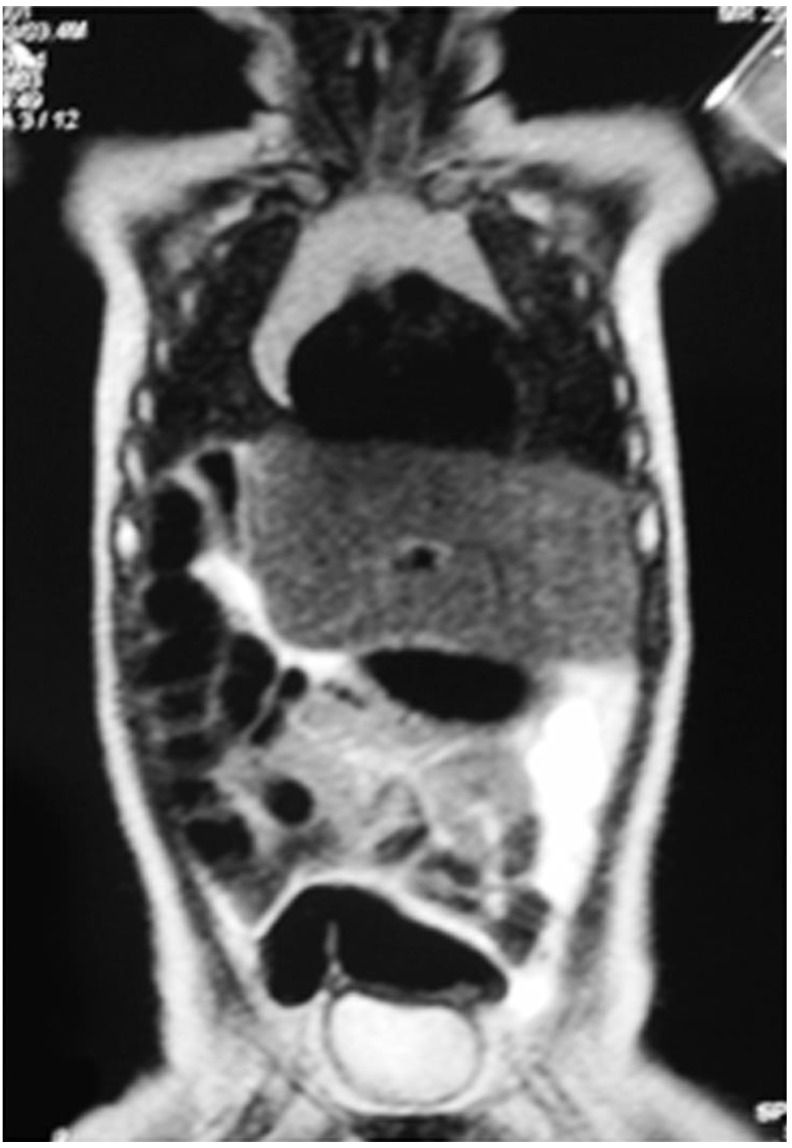
Total body MRI at 6 months follow-up showing no recurrence of disease.

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
