# Peer review of "Liver Resection Using Saline-Linked Radiofrequency Technology in an Infant with Congenital Hepatoblastoma"

_children, 2022, doi:10.3390/children9030418_

Round 1

Reviewer 1 Report

This manuscript reports on a case of giant congenital hepatoblastoma treated with neoadjuvant chemotherapy and hepatic resection in a 3-months-old male. The manuscript is well written and organized and contains results that could be interesting for some readers. There are some comments that should be taken into account before accepting for publication.

  • 1st page, in Introduction, 2nd paragraph is too short. It is not customary that one paragraph is composed of only one sentence.
  • 1st page, in Introduction, 3rd paragraph is too short. It is not customary that one paragraph is composed of only one sentence.  Maybe to put  2nd and 3rd paragraphs together.
  • 1st page, in Introduction, 6th paragraph is too short. It is not customary that one paragraph is composed of only one sentence.
  • 2nd page, in the Case report, the last 2 paragraphs should be put together.
  • 2nd page, I am not sure that Table 1 does not look more like Figure than the Table.
  • 6th page, Discussion is very good, but again the first paragraph is composed of only one paragraph.  
  • The list of references is ok for the Case Report. I am only surprised that one recent paper is not cited:

Radjenović, B. Et al., On Efficacy of Microwave Ablation in the Thermal Treatment of an Early-Stage Hepatocellular Carcinoma. Cancers 202113, 5784.

Author Response

Dear Reviewer,

Thanks a lot for your review and for your useful suggestions. We corrected the body of the manuscript by your indications. We added the reference you suggested. We deleted Table 1 by the indication of the reviewer 2.

Reviewer 2 Report

Reviewer’s comments

This is an interesting report of a rare case of neonatal hepatoblastoma. The authors have systematically presented the cases and reviewed the literature.

The author focuses on surgical procedures, but for the pediatricians, I think it would be better to describe diagnosis and anti-cancer drug treatment in a little more detail.

The paper is clearly written, however there are some points that need to be revised.

  1. Please describe the properties (size, position, properties) of the tumor in the fetal diagnosis, and if possible, present an image of fetal ultrasound. What kind of diseases were listed as differential diagnosis in fetal findings ?
  2. AFP is elevated in the normal neonates, so it is necessary to indicate the normal value of AFP in the newborn and describe whether it was higher than that.
  3. Table 1 is listed in the textbook, so I don't think it is necessary.
  4. Figure 1; Is the right one at the time of admission and the left one after chemotherapy? Please specify.
  5. Preoperative chemotherapy:Did you use the SIOPEL protocol? In newborns, the amount of anticancer drug is usually reduced, It should be stated how much was actually administered and how the dose was decided.
  6. Figure 2 It is not necessary to show the image that there is no recurrence. If the main focus is on the technical description of surgery, photos during surgery and photos of excised tumors should be presented.

Author Response

Dear reviewer,

Thanks for your suggestions! We revised the manuscript by your indications. We added figure 2 because unfortunately we haven't operative images due to a breakdown in our image capture system.